# 3D INTERACTING HANDS DIFFUSION MODEL

## ABSTRACT

Humans make two-hands interactions in a variety of ways. Learning prior distributions of interactions between hands is critical for 1) generating new interacting hands and 2) recovering plausible and accurate interacting hands. Unfortunately, there have been no attempts to learn the prior distribution of interactions between two hands. Due to the lack of prior distribution, previous 3D interacting hands recovery methods often produce hands with physically implausible interactions, such as severe collisions, and semantically meaningless interactions. We present IHDiff, the first generative model for learning the prior distribution of interacting hands. Motivated by the strong performance of recent diffusion models, we learn the prior distributions using the diffusion process. For the reverse diffusion process, we design a novel Transformer-based network, which effectively captures correlations between joints of two hands using self- and cross-attention. We showcase three applications of IHDiff including random sampling, conditional random sampling, and fitting to observations. The code and pre-trained model will be publicly available.

## 1 INTRODUCTION

Humans frequently engage in two-hand interactions when expressing emotions through hand gestures and interacting with objects. The complicated articulations of hands and diverse patterns of hand interactions make analyzing and understanding the interactions greatly challenging. Recent introduction of large-scale 3D interacting hands datasets (Moon et al., 2020; 2022) motivated many studies (Rong et al., 2021; Zhang et al., 2021; Li et al., 2022; Hampali et al., 2022; Di & Yu, 2022; Fan et al., 2021; Kim et al., 2021; Meng et al., 2022; Moon, 2023), which aim to recover 3D interacting hands from a single image.

Despite their great achievements, most of them have not tackled the problem of **generating** new 3D interacting hands by modeling prior distributions of interacting hands. Most of the existing works are based on a discriminative approach, which does not model a prior distribution of data. Some works Wang et al. (2022); Zuo et al. (2023) are based on a conditional generative approach to learn conditional distributions; however, theirs are conditioned on image features, which makes unconditional random sampling impossible. Such generation is greatly useful for creating animations of virtual humans, which can be combined with recent advancements of generative artificial intelligence (Ho et al., 2020; Song et al., 2021a; Song & Ermon, 2020; Song et al., 2021b; Rombach et al., 2022).

In addition to the generation, modeling prior distributions is also critical for **recovering 3D interacting hands in the wild**. For example, in most cases of the two-hand interactions, only partial hand joints are visible, which makes predictions from existing 3D interacting hands regressors (Rong et al., 2021; Zhang et al., 2021; Li et al., 2022; Hampali et al., 2022; Di & Yu, 2022; Fan et al., 2021; Kim et al., 2021; Meng et al., 2022; Moon, 2023) suffer from implausible outputs, such as collisions, and meaningless interactions without correct contact between two hands. The problem becomes more severe when it comes to the in-the-wild case due to a domain gap between in-the-lab training sets and in-the-wild testing sets. Usually, 3D annotations are available only for images captured from a lab environment, such as InterHand2.6M (Moon et al., 2020) dataset, where those images have very different appearances (*e.g.*, background, color space, and illuminations) from those of in-the-wild images. Due to such a large image appearance domain gap, simply training existing methods on large-scale datasets (Moon et al., 2020) is not enough to make systems robust to in-the-wild images. On the other hand, the prior distributions are not conditioned on any data

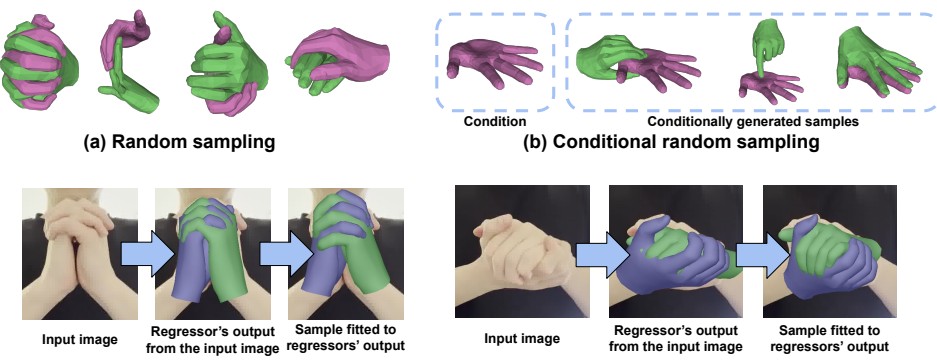

(a) Random sampling

Condition | Conditionally generated samples
(b) Conditional random sampling

Input image | Regressor's output from the input image | Sample fitted to regressors' output

Input image | Regressor's output from the input image | Sample fitted to regressors' output

(c) Fitting to observations

Figure 1: Three applications of our IHDiff. Using learned prior distribution, IHDiff enables (a) unconditional random sampling and (b) conditional sampling. In addition, (c) shows that predictions from off-the-shelf regressors (Moon, 2023) can be projected to our learned prior distribution for plausible 3D interacting hands. All three applications can be performed with a single IHDiff without additionally training application-specific generative models.

including images, which can be useful to reduce the domain gap between in-the-lab and in-the-wild environments. For example, physically implausible outputs of existing methods from in-the-wild images can become plausible and have meaningful interactions by being projected to the learned prior distributions.

We propose IHDiff, the first generative model to learn prior distributions of interactions between two hands in the 3D space. IHDiff models the prior distribution following the diffusion-based generative models (Ho et al., 2020; Song et al., 2021a; Song & Ermon, 2020; Song et al., 2021b; Rombach et al., 2022) due to their powerful capabilities. After gradually making the data to noise by the forward diffusion process, our IHDiff denoises the noised data to the original data using our novel Transformer (Vaswani et al., 2017)-based network, which effectively captures the correlation between joints of two hands using the self- and cross-attention. Instead of predicting added noise for the denoising like previous diffusion-based generative models (Ho et al., 2020; Song et al., 2021a; Song & Ermon, 2020; Song et al., 2021b), we design IHDiff to predict the original clean sample. Therefore, we can utilize various geometric losses, such as forward kinematics loss and collision avoidance loss, to the denoised clean sample during the training.

We showcase three applications of our IHDiff. Please note that all three applications can be done using *a single IHDiff* without any additional training. Fig. 1 shows examples of the three applications. First, we show that randomly sampled 3D interacting hands from our learned prior distribution have physically plausible, semantically meaningful, and diverse interactions. Second, we show conditional random sampling. For example, we generate a 3D left hand conditioned on the 3D right hand. Finally, we show that our IHDiff is effective in recovering plausible interacting hands given noisy and partially available observations, which could be used as post-processing of existing 3D interacting hands recovery methods (Rong et al., 2021; Zhang et al., 2021; Li et al., 2022; Hampali et al., 2022; Di & Yu, 2022; Fan et al., 2021; Kim et al., 2021; Meng et al., 2022; Moon, 2023).

## 2 RELATED WORKS

**3D interacting hands recovery.** Early works (Oikonomidis et al., 2012; Ballan et al., 2012; Tzionas et al., 2016; Taylor et al., 2016; Mueller et al., 2019; Wang et al., 2020) are based on a fitting framework, which fits 3D hand models to geometric observations, such as RGBD sequence (Oikonomidis et al., 2012), hand segmentation map (Mueller et al., 2019), and dense matching map (Wang et al., 2020). Recently, (Moon et al., 2020; 2022) presented the IH2.6M dataset, a large-scale and real-captured dataset that contains multi-view images with 3D annotations of interacting hands. Motivated by IH2.6M, several regression-based methods (Rong et al., 2021; Zhang et al., 2021; Li et al., 2022; Hampali et al., 2022; Di & Yu, 2022; Fan et al., 2021; Kim et al., 2021; Meng et al.,

2022; Moon, 2023) have been proposed, which perform better than the early fitting-based methods. (Zhang et al., 2021) proposed a sequentially refining 3D interacting hand mesh estimation system. (Li et al., 2022) proposed a Transformer-based network for accurate 3D interacting hand reconstruction. (Wang et al., 2022; Zuo et al., 2023) introduced a conditional generative model to learn conditional distributions, not prior distributions. However, as their condition is an image, they cannot generalize well to in-the-wild images due to the large image appearance domain gap. Moon (Moon, 2023) presented a 3D interacting hands recovery system that produces robust outputs from in-the-wild images.

None of the above approaches consider prior distributions of interactions between two hands. Hence, they often produce physically implausible interactions or semantically meaningless interactions between two hands, especially from in-the-wild images. In addition, they cannot be used for *generating* new 3D interacting hands, which is critical for creating animations of virtual humans. Our IHDiff is the first generative model that learns the prior distribution of interactions between two hands. Our learned prior distribution can be used for 1) making outputs of existing recovery methods plausible even for the in-the-wild case and 2) generating new 3D interacting hands.

**Generative models for 3D humans.** Early works modeled prior distributions of the 3D human body pose with a Gaussian mixture model (GMM). Such GMM-based prior is used for downstream tasks, such as 3D body pose fitting (Bogo et al., 2016). The principal component analysis is also widely used to model human prior distributions (Ormoneit et al., 2000; Romero et al., 2017). With the rise of deep learning, more sophisticated generative models have been introduced. VPoser (Pavlakos et al., 2019) is a variational autoencoder (VAE)-based generative model, which models prior distributions of the human body pose space. (Zanfir et al., 2020; Kolotouros et al., 2021) use the normalizing flow to recover 3D human body pose from a single image. (Davydov et al., 2022) introduces adversarial pose prior of the 3D human body. Pose-NDF (Tiwari et al., 2022) is a neural distance fields-based generative model. It treats each posed human body as a single point in the high-dimensional distance fields and learns the shortest distance from each point to the desired manifolds. In addition to the above pose priors, many generative models for learning human motion priors have been introduced. HuMoR (Rempe et al., 2021) models the prior distribution of human body motion using a conditional VAE. ACTOR (Petrovich et al., 2021) is a conditional VAE-based approach to generate a short-term 3D human motion from an action label. MultiAct (Lee et al., 2023) extends ACTOR (Petrovich et al., 2021) by generating an infinite length of 3D human motion from multiple action labels. NeMF (He et al., 2022) utilizes neural motion fields to generate 3D human motions. MDM (Tevet et al., 2023) is a diffusion-based generative model, which can be used for text-conditioned, action label-conditioned, or even unconditional 3D human body motion generation. Most of the above generative models are for learning priors of 3D human *body*. On the other hand, our IHDiff is the first generative model for learning priors of *interaction between two hands*.

## 3 IHDIFF

IHDiff consists of forward diffusion and reverse diffusion processes. The forward diffusion process takes the data $\boldsymbol{X}_0$ and gradually makes it to a Gaussian noise $\boldsymbol{X}_t$ following DDPM (Ho et al., 2020). Then, the reverse diffusion process denoises the Gaussian noise to the original data $\hat{\boldsymbol{X}}_0$.

### 3.1 FORWARD DIFFUSION

**Inputs.** We first prepare 1) 3D joint coordinates of the right hand $\boldsymbol{J}^{\mathrm{R}} \in \mathbb{R}^{J \times 3}$, 2) 3D joint coordinates of the left hand $\boldsymbol{J}^{\mathrm{L}} \in \mathbb{R}^{J \times 3}$, 3) 3D joint angles of the right hand $\boldsymbol{\Theta}^{\mathrm{R}} \in \mathbb{R}^{J \times 6}$, and 4) 3D joint angles of the left hand $\boldsymbol{\Theta}^{\mathrm{L}} \in \mathbb{R}^{J \times 6}$. The 3D coordinates include relative translation between two hands, necessary to represent a status of two hands (*e.g.*, contact between two hands). The 3D joint angles are represented as a 6D rotational representation (Zhou et al., 2019). $J = 21$ denotes the number of joints of a single hand. We normalize all inputs to the right hand-relative space by canceling 3D global rotation and 3D global translation of the right hand. Then, the four items are concatenated to a single matrix $\boldsymbol{X}_0 \in \mathbb{R}^{2J \times 9}$.

**Adding Gaussian noise.** Following DDPM (Ho et al., 2020), we gradually make the original data $\boldsymbol{X}_0$ to a Gaussian noise $\boldsymbol{X}_t$ like below.

$$\boldsymbol{X}_t = \sqrt{\bar{\alpha}_t}\boldsymbol{X}_0 + \sqrt{(1 - \bar{\alpha}_t)}\epsilon_t, \tag{1}$$

where $t$ denotes the noising step and is uniformly sampled from $[1, T]$. $T = 1000$ denotes the maximum noising step. $\bar{\alpha}_t = \Pi_{s=1}^t \alpha_s$ is a scalar constant, and we pre-define $\beta_t = 1 - \alpha_t$ using a cosine scheduler (Nichol & Dhariwal, 2021). $\epsilon_t$ is a random normal Gaussian noise, which has the same dimension of $\boldsymbol{X}_0$. The forward diffusion process is not learnable, and it is performed with pre-defined constants $\{\alpha_t\}_{t=1}^T$.

## 3.2 REVERSE DIFFUSION

Unlike the forward diffusion process, not a learnable process, the reverse diffusion process is driven by a learnable neural network. To this end, we design a novel Transformer (Vaswani et al., 2017)-based network $f$.

**Novel Transformer-based denoising network $f$.** Fig. 2 shows the pipeline of the denoising network $f$. It takes the noised data $\boldsymbol{X}_t$ and the noising step $t$ as inputs. Then, it outputs original data $\hat{\boldsymbol{X}}_0$ like below.

$$\hat{\boldsymbol{X}}_0 = f(\boldsymbol{X}_t, t). \tag{2}$$

The noisy 3D coordinate and 6D rotation in $\boldsymbol{X}_t$ from the same handedness are passed to a shared linear layer to embed them from $3 + 6$ dimension to $c = 256$ dimension. We also embed the noising step $t$ to $c$-dimensional space by changing it to the trigonometric positional encoding and passing it to multi-layer perceptron (MLP) with SiLU activation function (Elfwing et al., 2018). Then, the embedded noising step, 3D coordinates, and 6D rotations of the same handedness are concatenated, which results in $1 + J$ tokens for each hand, where each token is a $c$-dimensional vector.

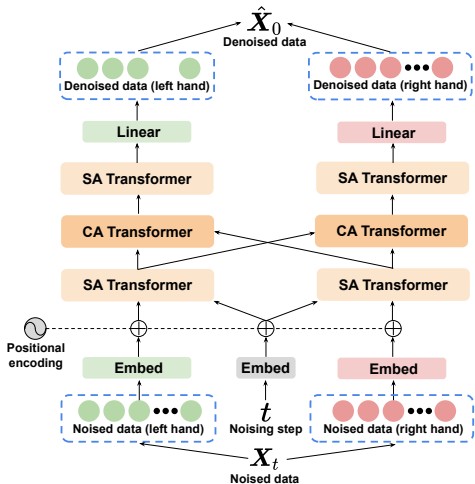

Figure 2: The pipeline of the denoising network $f$, used in the reverse diffusion process of IHDiff.

After adding a learnable positional encoding, we first pass $1 + J$ tokens of each hand to separate the self-attention (SA) Transformer to model dependencies between joints within each right and left hand. To consider interactions between two hands, we pass the left hand outputs as queries and right hand outputs as keys and values to a cross-attention (CA) Transformer, which results in left hand tokens, enhanced by considering relationships to the right hand tokens. In the same manner, we obtain right hand tokens, enhanced by the left hand tokens, with another CA Transformer. We found that such CA Transformers are necessary to model interactions between two hands, which makes our denoising network $f$ distinctive from existing 3D human body modeling works Tevet et al. (2023); Xin et al. (2023). Finally, we again pass tokens of each hand to separate the SA Transformer to further enhance features based on information of the other hand, captured by the previous CA Transformer.

After obtaining the final outputs, we discard an output from the noising step token. The remaining outputs from the same handedness are passed to a shared linear layer to restore original data, which produces $\hat{\boldsymbol{X}}_0$. $\hat{\boldsymbol{X}}_0$ is splitted to restored 1) 3D joint coordinates of the right hand $\hat{\boldsymbol{J}}^{\text{R}}$, 2) 3D joint coordinates of the left hand $\hat{\boldsymbol{J}}^{\text{L}}$, 3) 3D joint angles of the right hand $\hat{\boldsymbol{\Theta}}^{\text{R}}$, and 4) 3D joint angles of the left hand $\hat{\boldsymbol{\Theta}}^{\text{L}}$.

**Shape parameter estimator.** In addition to restoring the original input, we estimate both hands' shape parameters of MANO hand model (Romero et al., 2017), denoted by $\hat{\gamma}^{\text{R}}$ and $\hat{\gamma}^{\text{L}}$, from the restored $\hat{\boldsymbol{X}}_0$ using an MLP with two fully-connected layers and the ReLU activation function. Another design choice is providing the MANO shape parameter as a conditional signal of the denoising network $f$; however, we found that our denoising network $f$ tries to memorize the corresponding status of interacting hands from the MANO shape parameter. To avoid such an undesired memorization, we design a separate shape parameter estimator.

**Final outputs.** We pass $\hat{\Theta}^{\mathrm{R}}$ and $\hat{\gamma}^{\mathrm{R}}$ to the MANO layer to obtain 3D mesh of the right hand $\hat{V}^{\mathrm{R}}$. Likewise, we pass $\hat{\Theta}^{\mathrm{L}}$ and $\hat{\gamma}^{\mathrm{L}}$ to the MANO layer to obtain 3D mesh of the left hand $\hat{V}^{\mathrm{L}}$. $\hat{V}^{\mathrm{R}}$ is translated to have the same root joint (*i.e.*, wrist) coordinate as $\hat{J}^{\mathrm{R}}$. $\hat{V}^{\mathrm{L}}$ is translated in the same way using $\hat{J}^{\mathrm{L}}$. The final translated $\hat{V}^{\mathrm{R}}$ and $\hat{V}^{\mathrm{L}}$ are the final output of the reverse diffusion process.

### 3.3 Loss functions

We train the denoising network $f$ by minimizing $L$, defined below.

$$L = L_{\boldsymbol{J}} + L_{\boldsymbol{\Theta}} + L_{\boldsymbol{V}} + L_{\gamma} + L_{\mathrm{col}}, \tag{3}$$

$$L_{\boldsymbol{J}} = ||\boldsymbol{J}^{\mathrm{R}} - \hat{\boldsymbol{J}}^{\mathrm{R}}||_1 + ||\boldsymbol{J}^{\mathrm{L}} - \hat{\boldsymbol{J}}^{\mathrm{L}}||_1, \quad L_{\boldsymbol{\Theta}} = ||\boldsymbol{\Theta}^{\mathrm{R}} - \hat{\boldsymbol{\Theta}}^{\mathrm{R}}||_1 + ||\boldsymbol{\Theta}^{\mathrm{L}} - \hat{\boldsymbol{\Theta}}^{\mathrm{L}}||_1, \tag{4}$$

$$L_{\boldsymbol{V}} = ||\boldsymbol{V}^{\mathrm{R}} - \hat{\boldsymbol{V}}^{\mathrm{R}}||_1 + ||\boldsymbol{V}^{\mathrm{L}} - \hat{\boldsymbol{V}}^{\mathrm{L}}||_1, \quad L_{\gamma} = ||\gamma^{\mathrm{R}} - \hat{\gamma}^{\mathrm{R}}||_1 + ||\gamma^{\mathrm{L}} - \hat{\gamma}^{\mathrm{L}}||_1, \tag{5}$$

where $\boldsymbol{V}^{\mathrm{R}}$ and $\boldsymbol{V}^{\mathrm{L}}$ denote original 3D meshes of the right and left hands, respectively. We found that $L_{\boldsymbol{V}}$ effectively prevents 3D joint angles' error accumulations along the kinematic chain. $\gamma^{\mathrm{R}}$ and $\gamma^{\mathrm{L}}$ denote the original MANO shape parameters of the right and left hands, respectively. $L_{\mathrm{col}}$ is a collision avoidance loss, computed by shooting a ray from each mesh vertex and checking the collision status and depth. Please refer to Sec. D for detailed descriptions of our $L_{\mathrm{col}}$. Ours $L_{\mathrm{col}}$ can prevent both collisions within each hand (*i.e.*, self-collisions) and between hands (*i.e.*, inter-collisions), while the widely used signed distance field (SDF)-based collision avoidance loss function (Rong et al., 2021) can only handle the inter-collisions. We show that our $L_{\mathrm{col}}$ is necessary to generate collision-free interacting hands in the Sec. 5.2, which has not been considered in existing 3D human body modeling works Tevet et al. (2023); Xin et al. (2023).

## 4 Applications

After training our reversing network $f$, our IHDiff can be used for various useful applications. Please note that all below three applications can be performed with a single IHDiff without any additional training.

---

**Algorithm 1** Sampling

1: **for** $n = N$ to 1 **do**
2:     $\hat{\boldsymbol{X}}_0 \leftarrow f(\boldsymbol{X}_n, n)$
3:     $\epsilon_n \leftarrow \frac{1}{\sqrt{1-\bar{\alpha}_n}}(\boldsymbol{X}_n - \sqrt{\bar{\alpha}_n}\hat{\boldsymbol{X}}_0)$
4:     $\boldsymbol{X}_{n-1} \leftarrow \sqrt{\bar{\alpha}_{n-1}}\hat{\boldsymbol{X}}_0 + \sqrt{1-\bar{\alpha}_{n-1}}\epsilon_n$
5: **end for**
6: **return** $\hat{\boldsymbol{X}}_0$

---

**Algorithm 2** Fitting

1: **for** $n = N$ to 1 **do**
2:     $\hat{\boldsymbol{X}}_0 \leftarrow f(\boldsymbol{X}_n, n)$
3:     $\epsilon_n \leftarrow \frac{1}{\sqrt{1-\bar{\alpha}_n}}(\boldsymbol{X}_n - \sqrt{\bar{\alpha}_n}\hat{\boldsymbol{X}}_0)$
4:     $\boldsymbol{X}'_{n-1} \leftarrow \sqrt{\bar{\alpha}_{n-1}}\hat{\boldsymbol{X}}_0 + \sqrt{1-\bar{\alpha}_{n-1}}\epsilon_n$
5:     $\boldsymbol{X}_{n-1} \leftarrow \boldsymbol{X}'_{n-1} - \nabla_{\boldsymbol{X}_n} g(\hat{\boldsymbol{X}}_0, \boldsymbol{Y})$
6: **end for**
7: **return** $\hat{\boldsymbol{X}}_0$

---

### 4.1 Random sampling

To randomly sample 3D interacting hands from our learned prior distributions, we first initialize $\boldsymbol{X}_N$ as a random normal Gaussian matrix. Then, we use DDIM (Song et al., 2021a) to efficiently restore 3D interacting hands from $\boldsymbol{X}_N$. Alg. 1 describes the sampling process. We use $N$ steps for the DDIM. Please note that the above random sampling is an *unconditional* random sampling.

### 4.2 Conditional random sampling

For the conditional random sampling, we first initialize $\boldsymbol{X}_N$ as a random normal Gaussian matrix. Then, we follow Alg. 1 for the random sampling. The only difference from the unconditional random sampling of Sec. 4.1 is that we simply overwrite a part of $\hat{\boldsymbol{X}}_0$ with conditional data in Alg. 1 for every iteration. In this way, we can encourage the sampling process to generate the remaining parts by considering overwritten conditional data.

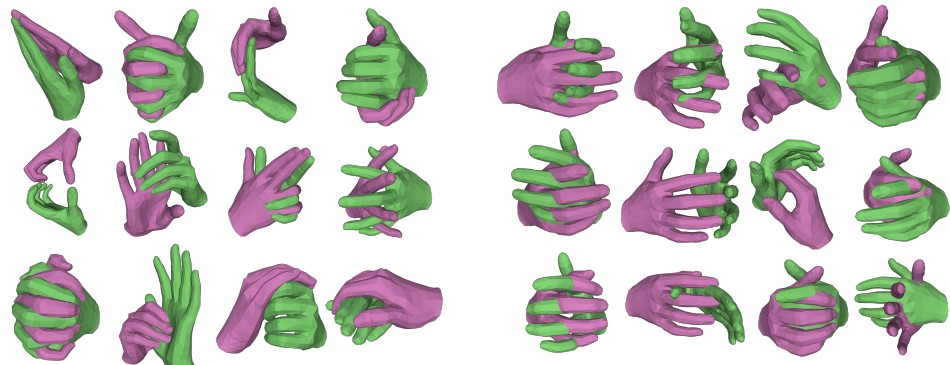

(a) Randomly sampled hands using IHDiff (Ours)    (b) Randomly sampled hands using the VAE baseline

Figure 3: Qualitative comparison between randomly generated samples from (a) our IHDiff and (b) the VAE baseline.

### 4.3 FITTING TO OBSERVATIONS

We first initialize $X_N$ as a random normal Gaussian matrix. If regression results from off-the-shelf regressors are available, we obtain $X_N$ by making the regression results to a Gaussian noise following Eq. 1. Then, we fit $X_N$ to given observations to obtain 3D interacting hands corresponding to the observations. There could be various types of observations, such as 2D hand joint coordinates, 3D hand joint coordinates, and 3D hand mesh vertices. Alg. 2 shows the fitting process, similar to (Chung et al., 2023). $Y$ and $g$ in Alg. 2 denote the observations and a loss function for the fitting, respectively.

## 5 EXPERIMENTS

### 5.1 EXPERIMENT PROTOCOLS

**Datasets.** We train our denoising network $f$ on the training split of the 30 fps version of IH2.6M dataset (Moon et al., 2020) and Re:InterHand Moon et al. (2023). Then, we test our IHDiff on the test split of the IH2.6M dataset. To learn meaningful interaction between two hands, we use samples only when the shortest distance between two-hands' mesh vertices is shorter than 3 mm.

**Baselines.** As there is no generative model for 3D interacting hands, we implemented a VAE baseline, which has a similar architecture of state-of-the-art unconditional 3D human motion generator (Xin et al., 2023; Petrovich et al., 2021), and compare it with our IHDiff. As the original VAE-based network is for the 3D human body motion generation, we modified it to a 3D interacting hand generator by changing its per-timestep tokens to per-joint tokens like our network $f$. We set the dimension of the baseline's learnable tokens to $(7, 256)$ following (Xin et al., 2023). We found that the above VAE baseline produces better samples than a V-Poser-style Pavlakos et al. (2019) VAE. For more details on the VAE baseline and descriptions of other possible baselines (Tiwari et al., 2022), please refer to Sec. E.1 and Sec. E.2, respectively.

Table 1: Quantitative comparison between the VAE baseline and our IHDiff using randomly sampled hands.

| Methods | APD (mm) | Col. (%) |
|---|---|---|
| VAE | 28.44 | 3.71 |
| IHDiff (wo. $L_{col}$) | 56.11 | 2.58 |
| **IHDiff (Ours)** | **56.23** | **1.70** |

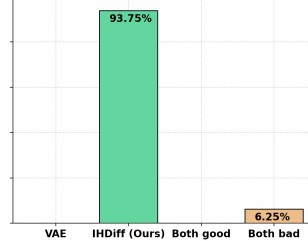

Figure 4: User study with randomly sampled hands from the VAE baseline and IHDiff.

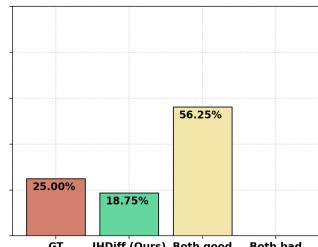

Figure 5: User study with GT and randomly sampled hands from our IHDiff.

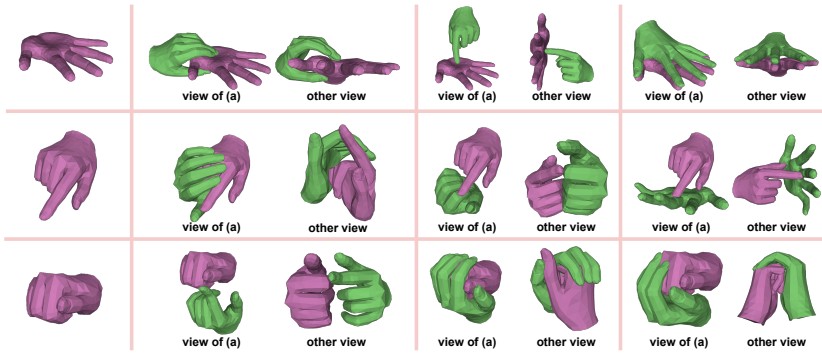

(a) Conditional right hand      (b) Randomly sampled hands conditioned on (a)

Figure 6: Qualitative results of conditionally generated samples using our IHDiff.

## 5.2 RANDOM SAMPLING

Fig. 3 shows that randomly sampled 3D interacting hands from our IHDiff are much more realistic and diverse than those from the VAE baseline. The random sampling is conducted following descriptions of Sec. 4.1. Most of the samples from the VAE baseline suffer from severe collisions. In addition, their interaction patterns are monotonous; for example, in most cases, the right hand has a simple neutral pose, and the two hands face each other. On the other hand, samples from our IHDiff suffer much less from collisions, while having diverse interaction patterns.

Tab. 1 shows that our IHDiff achieves higher average pairwise distance (APD), an average distance between all pairs of generated 3D interacting hands' meshes. This indicates that ours produces more diverse samples. The table also shows that randomly sampled ones from IHDiff suffer less from the collision. Interestingly, the table shows that without our collision avoidance loss $L_{col}$, the ratio of colliding vertices increases by about 52%, which shows the effectiveness of our $L_{col}$. The numbers in the table are computed from randomly sampled 512 3D interacting hands.

Finally, we conducted user studies to further qualitatively compare 1) VAE baseline and IHDiff in Fig. 4 and 2) groundtruth (GT) from the IH2.6M dataset and IHDiff in Fig. 5. The user studies clearly show the superiority of IHDiff over the VAE baseline. Interestingly, our IHDiff achieves comparable results compared to GT. For each user study, we asked 16 questions to 33 users, where each question shows two side-by-side videos. The videos include rendered 3D interacting hands from rotating viewpoints of comparing methods. We let users select a video that contains more realistic 3D interacting hands. We also provided 'similarly good' and 'similarly bad' options. Please refer to Sec. F for more details and screenshots of our user study.

## 5.3 CONDITIONAL RANDOM SAMPLING

Fig. 6 shows conditional random sampling examples. As described in Sec. 4.2, we first randomly generate 3D interacting hands. Then, we pick the right hand (Fig. 6 (a)) and use it as conditional data for the conditional random sampling. Fig. 6 (b) shows that IHDiff is able to generate diverse and semantically meaningful interactions although it is trained only in an unconditional generation pipeline. Such a simple extension to conditional data generation is not possible for the VAE baseline as it requires another conditional VAE to be trained.

## 5.4 FITTING TO OBSERVATIONS

In this subsection, we report three metrics: **vertex error**, **ratio of colliding vertices**, and **contact accuracy**. The vertex error (the lower the better) is a Euclidean distance between recovered and GT 3D meshes. The ratio of colliding vertices (the lower the better) is the ratio of vertices whose collision depth is bigger than 3 mm. Finally, the contact accuracy (the higher the better) is the ratio of correct contact among vertices that are annotated as contacting. Please note that *we first introduce and report the contact accuracy between two hands*, which is greatly important to describe interactions between two hands.

Table 2: Quantitative comparison of nearest neighbor (NN) search, VAE baseline, and IHDiff on various test sets. For each evaluation, the number indicates the vertex error (mm) / ratio of collision (%) / contact accuracy (%).

| Methods | Test sets | | | |
| --- | --- | --- | --- | --- |
| | Clean | Jitter | Swap | Partial |
| NN search | 20.97 / **1.04** / 10.92 | 22.89 / **1.40** / 11.02 | 23.32 / **0.98** / 10.69 | 28.66 / **1.12** / 9.71 |
| VAE | 16.79 / 1.50 / 9.17 | 19.11 / 1.54 / 7.01 | 20.42 / 1.68 / 9.08 | 19.30 / 1.49 / 7.99 |
| IHDiff (SA only) | 6.55 / 1.50 / 24.42 | 10.06 / 1.50 / 12.48 | 9.58 / 1.66 / 22.84 | 8.20 / 1.48 / 22.53 |
| **IHDiff (Ours)** | **6.13** / 1.44 / **30.67** | **8.38** / 1.51 / **15.19** | **7.13** / 1.58 / **28.05** | **6.41** / 1.43 / **27.94** |

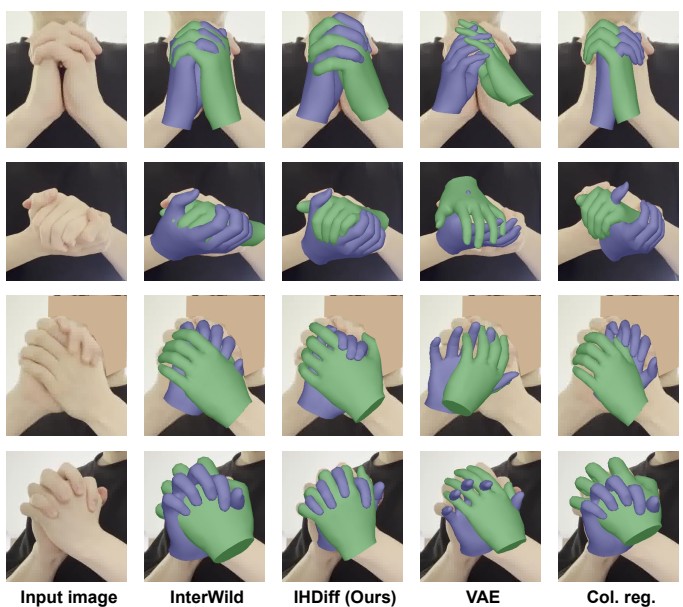

**Input image**    **InterWild**    **IHDiff (Ours)**    **VAE**    **Col. reg.**

Figure 7: Qualitative comparison of the fittings on in-the-wild images.

**Simulated noisy targets.** Tab. 2 shows that our IHDiff achieves the lowest vertex errors and highest contact accuracy on various types of test sets compared to the NN search and VAE baseline. It also achieves a comparable collision metric compared to the NN search, which retrieves the nearest GT samples. The 'Clean' test set is the IH2.6M test set without any noise. To check the robustness to noisy fitting targets, we added three types of noise to the 'Clean' test set. The 'Jitter' test set is made by adding a random Gaussian noise with a 10 mm standard deviation to the 'Clean' test set for all joints. The 'Swap' test set is made by changing random two joints from the right and left hands. The 'Partial' test set is made by removing four random joints of two hands. 'Swap' and 'Partial' simulate wrong-fitting targets due to the self-similarity between two hands and occlusions, respectively. For each test set, we applied exactly the same noise for all comparing methods to remove randomness.

The VAE baseline and IHDiff consistently achieve lower vertex error than the NN search (Johnson et al., 2019), which searches nearest examples from 1) a training set of IH2.6M and 2) our newly captured dataset. In particular, our IHDiff achieves better than half of the vertex errors for noisy targets compared to the VAE baseline. The table additionally shows that IHDiff only with SA performs much worse than our IHDiff, which consists of a combination of SA and CA. The variant only with SA has SA Transformers that take all $1 + 2J$ tokens of two hands, which have a similar number of parameters to our SA and CA Transformers, to capture intra- and inter-hand dependencies at the same time. We think the variant only with SA could suffer from the undesired correlation between left hand and right hand because it always computes the attention map from tokens of both left and right hands. On the other hand, we design the separate SA Transformers for left and right hands to effectively addresses the problem, which results in better locality of tokens and generalizability to unseen data, while capturing the dependency between two hands with the CA Transformer. Such results show the necessity of using a combination of SA and CA, which makes

Table 3: Quantitative comparison of various methods on HIC dataset (Tzionas et al., 2016).

| Methods | Vertex error (mm. ↓) | Collision (%. ↓) | Contact (%. ↑) |
|---|---|---|---|
| InterWild | **20.14** | 1.15 | 6.99 |
| InterWild + Col. reg. | 21.44 | 0.90 | 5.35 |
| InterWild + VAE | 59.88 | 3.87 | 3.38 |
| **InterWild + IHDiff (Ours)** | 20.89 | **0.86** | **12.89** |

ours distinctive from existing 3D human body modeling works Tevet et al. (2023); Xin et al. (2023), which only rely on SA to model the 3D motion of a single person. Our high contact accuracy is also noticeable as contact between two hands is one of the major factors in defining interaction patterns between two hands. We consider a vertex to be contacting if the shortest distance from that vertex to vertices of the other hand is smaller than 3 mm.

For all fittings, 3D joint coordinates are the fitting targets, and a distance between 1) target 3D joint coordinates and 2) 3D joint coordinates from the mesh is minimized during the fitting. After the fitting, the errors in the table are computed between output mesh vertices and GT mesh vertices after aligning the translation of the right hand. For the VAE baseline and our IHDiff, we start the fitting from latent samples, randomly initialized in normal Gaussian space. We followed Sec. 4.3 for the fitting of IHDiff. Please refer to Sec. B.1 for detailed descriptions of the fitting.

**Real-world noisy targets.** Fig. 7 shows that our IHDiff produces plausible 3D interacting hands from challenging in-the-wild images, while all comparing methods fail to. The images are newly captured with iPhone 13 from an indoor environment for this experiment. The results of Inter-Wild (Moon, 2023) are its regressor's outputs from the input images. We used InterWild as it 1) is a state-of-the-art regressor and 2) can detect hand boxes, while most of the other regressors do not work robustly on in-the-wild images and assume GT hand boxes. The results of the IHDiff and VAE baseline are obtained by fitting their latent samples to the output of InterWild. The initial latent sample of IHDiff is obtained by Eq. 1 using InterWild's outputs, and that of the VAE baseline is obtained by forwarding InterWild's output to its encoder. We followed Sec. 4.3 for the fitting of IHDiff. The results of Col. reg. are obtained by optimizing InterWild's outputs with collision avoidance regularizer while keeping them from being too far from InterWild's outputs, similar to (Rong et al., 2021).

Tab. 3 shows that combining our IHDiff with InterWild (Moon, 2023) produces the comparable vertex error, the lowest ratio of colliding vertices, and the highest contact accuracy on the HIC dataset (Tzionas et al., 2016). Please note that high contact accuracy is hard to achieve for InterWild and Col. reg. as they do not consider distributions of semantically meaningful interactions between two hands, where contact is a critical factor for such interaction semantics. For example, Fig. 7 shows that Col. reg. naively moves geometry to resolve the collisions without considering semantics of the hand interactions. We used samples from the HIC dataset only if the minimum distance between two-hand meshes is smaller than 3 mm. We found that as images of IH2.6M have similar appearances, existing regressors, trained on IH2.6M, are robust to IH2.6M's test set. Instead, we test the regressor on the HIC dataset as it contains indoor images, closer to the in-the-wild environment. Please refer to Sec. B.2 for detailed descriptions of the fitting.

## 6 CONCLUSION

**Summary.** We present IHDiff, the first generative model to learn prior distributions of the interaction between two hands. It is based on diffusion models, and for effective reverse diffusion, we introduce a novel Transformer (Vaswani et al., 2017)-based denoising network. We showcase three applications, including random sampling, conditional random sampling, and fitting.

**Limitations and future works.** IHDiff only considers kinematic-level interactions without elasticity. Therefore, there could be small collisions between hand meshes. Also, IHDiff does not model non-contacting two-hand interactions. Future work should incorporate our IHDiff with a neural network-based regressor for real-time 3D interacting hands recovery from a single image while enjoying our learned prior distributions.

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

## A  QUALITATIVE RESULTS

### A.1  RANDOM SAMPLING

Fig. 8 shows that our IHDiff generates diverse and realistic 3D interacting hands.

### A.2  CONDITIONAL RANDOM SAMPLING

Fig. 9 shows conditionally generated samples with conditional right hands (top two rows) and conditional left hands (bottom two rows). For the bottom two rows, in addition to the 3D pose of the left hand, 3D relative rotation and translation between the right and left hands are also included in the conditional left hand. Hence, all samples of each row from the bottom two rows have the same 3D relative rotation and translation between the two hands.

### A.3  FITTING TO OBSERVATIONS

**Simulated noisy targets.** Fig. 10 and  11 show that our IHDiff is robust to the 'Swap' and 'Partial' noise, respectively.

**Real-world noisy targets.** Fig. 12 shows that our IHDiff is greatly useful to obtain plausible 3D interacting hands from noisy estimations from off-the-shelf regressors Moon (2023). It is interesting that in addition to simply resolving colliding hands, our IHDiff produces plausible or semantically meaningful 3D interacting hands, as shown in the third row of the figure.

## B  FITTING

Sec. 5.4 shows two fitting scenarios. We provide details of each fitting scenario.

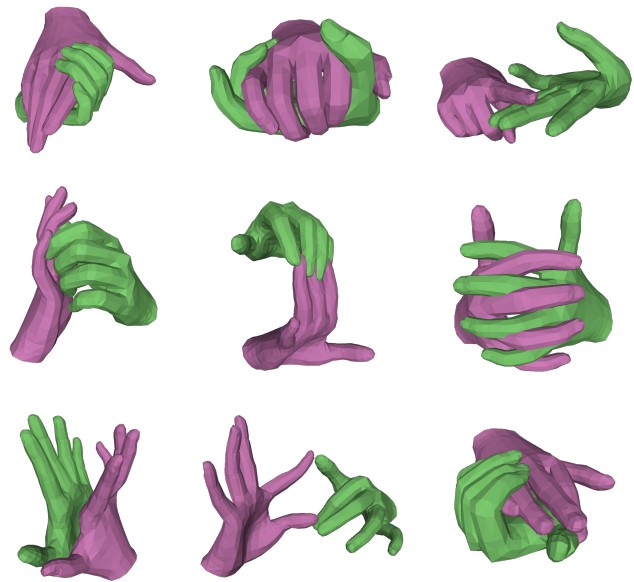

Figure 8: Qualitative results of randomly generated samples using our IHDiff.

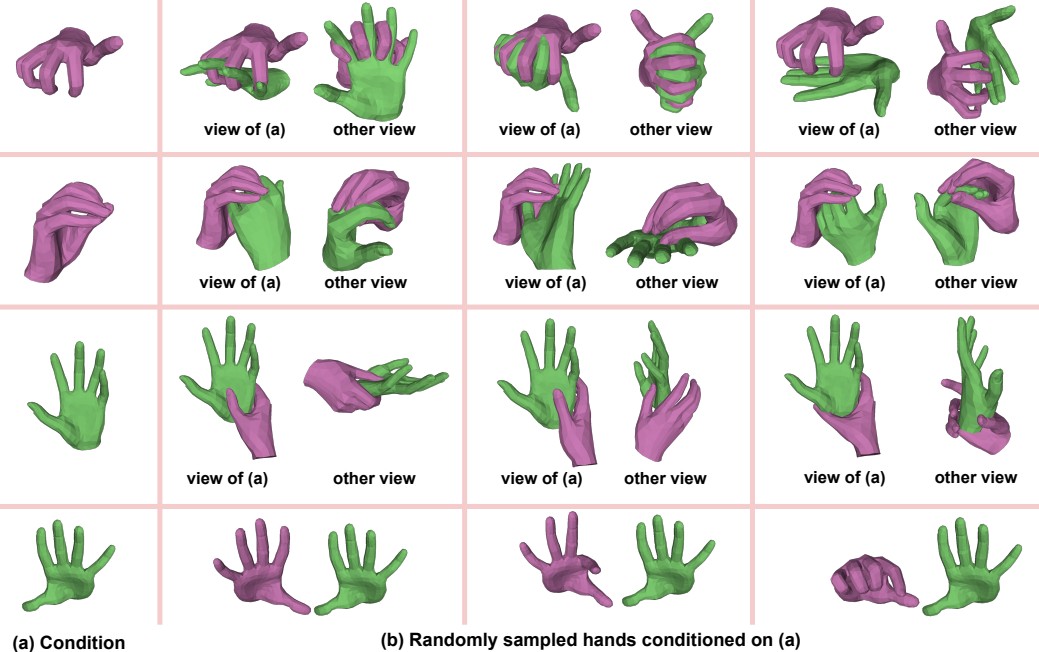

Figure 9: Qualitative results of conditionally generated samples using our IHDiff.

## B.1 SIMULATED NOISY TARGETS.

**IHDiff.** We use a weighted $L1$ distance as a loss function $g$ in Alg. 2. The weight of the $L1$ distance is set to 1000. For the best fitting result, we set the sampling iteration of DDIM $N$ to 1000. The update step is initially set to 1 and is decreased to 0.1 after 750 sampling iterations.

**VAE.** We use exactly the same weighted loss function of IHDiff for the VAE fitting. The fitting iteration is set to 1000. The update step is initially set to 0.01 and is decreased to 0.001 after 750 sampling iterations. We found that a high update step like 1 of IHDiff works badly for the VAE.

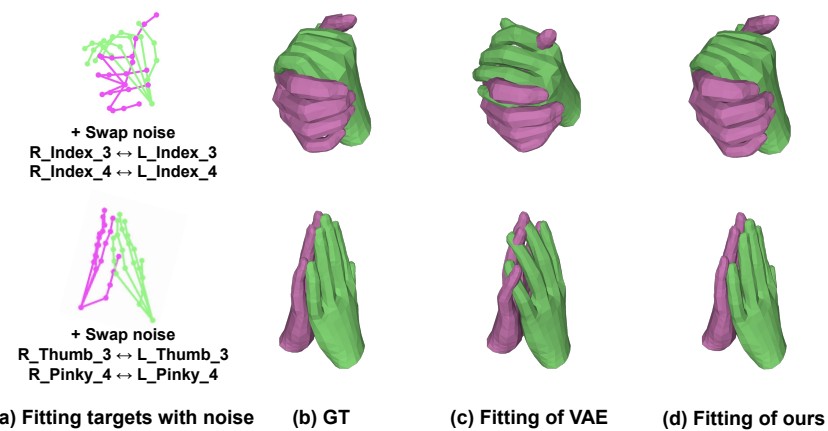

Figure 10: Qualitative comparison of the (b) GT, (c) fitting of the VAE baseline, and (d) fitting of our IHDiff. Fitting targets of (a) are made by corrupting GT 3D joint coordinates in the figure with a *swap* noise. It changes the coordinates of two joints. For example, 'R_Index_3 ↔ L_Index_3' means changing the right and left hands' third joints in the index finger. For all fingers, the direction from the first joint to the fourth joint starts from the finger root to the fingertip.

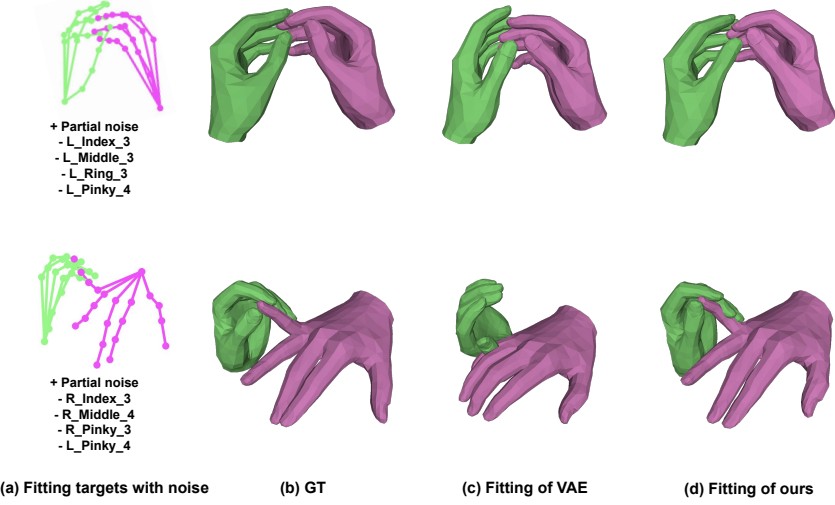

Figure 11: Qualitative comparison of the (b) GT, (c) fitting of the VAE baseline, and (d) fitting of our IHDiff. Fitting targets of (a) are made by corrupting GT 3D joint coordinates in the figure with a *swap* noise. It changes the coordinates of two joints. For example, 'R_Index_3 ↔ L_Index_3' means changing the right and left hands' third joints in the index finger. For all fingers, the direction from the first joint to the fourth joint starts from the finger root to the fingertip.

## B.2 REAL-WORLD NOISY TARGETS.

**IHDiff.** The loss function $g$ in Alg. 2 is a weighted sum of three loss functions. First, $L1$ distance between meshes from InterWild and the denoising network $f$. Second, $L1$ distance between the 3D pose from InterWild and the denoising network $f$. Finally, $L1$ distance between 2D hand joint coordinates from InterWild and the denoising network $f$. In addition to the latent sample from the learned prior distribution, we additionally optimize 3D global rotation and 3D global translation of the right hand as the right hand from IHDiff has normalized 3D global rotation and translation. To project the output of the denoising network $f$ to the 2D space, we additionally optimize orthogonal camera parameters. For the best fitting result, we set the sampling iteration of DDIM $N$ to 1000. The update step of the latent samples is initially set to 10 and is decreased to 1 after 750 sampling

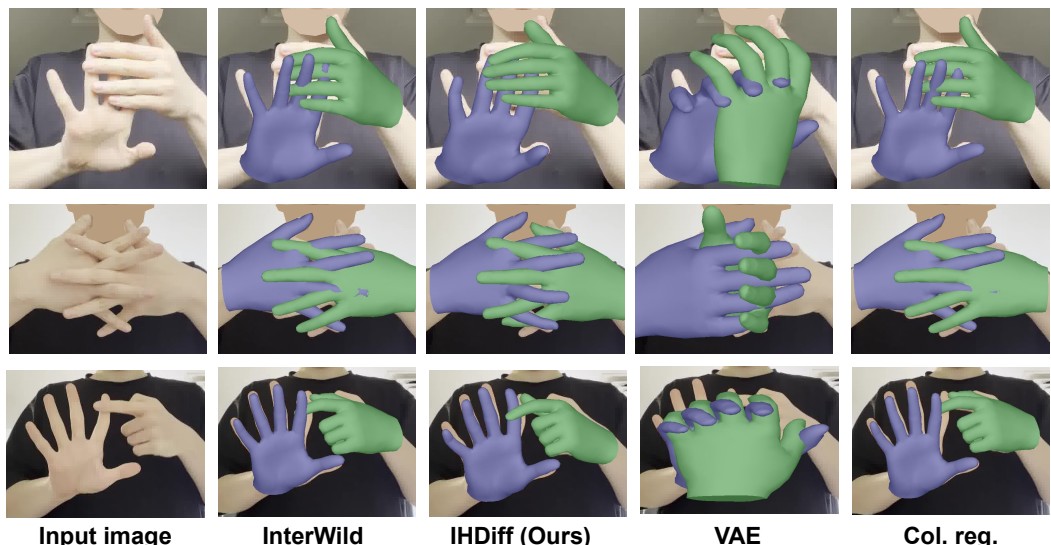

Figure 12: Qualitative comparison of the fittings on in-the-wild images.

iterations. On the other hand, the update step of the 3D global rotation, 3D global translation, and orthogonal camera parameters is initially set to 0.01 and is decreased to 0.001 after 750 sampling iterations.

**VAE.** We used exactly the same weighted loss functions as IHDiff. Like the fitting of IHDiff, 3D global rotation, 3D global translation, and orthogonal camera parameters are additionally optimized. The fitting iteration is set to 1000. The update step is initially set to 0.01 and is decreased to 0.001 after 750 sampling iterations.

**Col. reg.** Instead of fitting latent samples from prior distribution of generative models like IHDiff and VAE, we directly optimize InterWild's output 3D pose and 3D relative translation between two hands. We used exactly the same weighted loss functions as IHDiff. The update step is initially set to 0.01 and is decreased to 0.001 after 750 sampling iterations. We set the weight of the collision avoidance loss to 0.1 as larger than this weight makes severe artifacts, such as implausible 3D hand poses.

## C DENOISING NETWORK $f$

The denoising network $f$ consists of 4 linear layers, 1 MLP, 4 SA Transformers, and 2 CA Transformers. The linear layers embed the input signal or produce final outputs, and the MLP is to embed the noising step. All Transformers have a feature dimension of 256 with 4 heads. We use GeLU activation function Hendrycks & Gimpel (2016) in the Transformer.

The denoising network is trained with a single A100 GPU, which takes 1 day for the training. We initially set the learning rate to $10^{-4}$, and it decays by 0.1 at 85[th] and 95[th] epochs. The training is finished after 100 epochs. During the training, we set the size of the mini-batch to 32. We set the weight of the collision avoidance loss $L_{col}$ to 0.1.

## D COLLISION AVOIDANCE LOSS $L_{COL}$

Before computing the loss function, we fill a hole around the wrist of each left/right hand mesh by making a virtual vertex at the center of the wrist and faces around it to make 3D meshes from MANO a closed surface. Then, we make meshes of right and left hands into a single mesh by 1) concatenating vertices and faces and 2) shifting the later half of the concatenated faces (which

include vertex indices) by the number of single hand vertices (the number of original single-hand vertices + 1 due to the virtual vertex to fill the hole). Finally, we repeat the below procedures for each vertex $\mathbf{v}$ of the concatenated mesh using our custom CUDA implementation, which takes 0.0078 seconds for all vertices. This running time is almost the same as that of (Rong et al., 2021), which takes 0.0060 seconds but cannot detect the self-collisions.

1. We check whether a vertex $\mathbf{v}$ is inside of a mesh. To this end, we shoot a ray along its positive normal direction and count the number of intersections with faces using a ray tracing algorithm of Embree. If the number of the intersection is even, this vertex is not inside of a mesh; hence, return zero loss and terminate the loss computation.

2. From a vertex $\mathbf{v}$, we shoot a ray along its negative normal direction. Then, we get a face index and barycentric coordinates from the first intersecting face using a ray tracing algorithm of Embree, where the face index and barycentric coordinates represent potential target position $\mathbf{t}$ of the vertex $\mathbf{v}$ to resolve the collision. If there is no intersecting face, go to 3.

3. We check two conditions whether $\mathbf{t}$ is a target position of the vertex $\mathbf{v}$ to resolve the collision. First, the dot product between 1) a vector from $\mathbf{v}$ to $\mathbf{t}$ and 2) the normal of the first intersecting face should be positive. Second, the dot product between 1) a vector from $\mathbf{v}$ to $\mathbf{t}$ and 2) the normal of all $\mathbf{v}$'s neighbor faces should be negative. The second condition is to remove invalid shallow collisions to $\mathbf{v}$'s neighbor faces. If the two conditions are satisfied, go to 5.

4. From a vertex $\mathbf{v}$, we shoot a ray along its positive normal direction. Then, we get a face index and barycentric coordinates from the first intersecting face using a ray tracing algorithm of Embree, where the face index and barycentric coordinates represent potential target position $\mathbf{t}$ of the vertex $\mathbf{v}$ to resolve the collision. If there is no intersecting face, no collision is detected; hence, return zero loss and terminate the loss computation.

5. We check two conditions whether $\mathbf{t}$ is a target position of the vertex $\mathbf{v}$ to resolve the collision. First, the dot product between 1) a vector from $\mathbf{v}$ to $\mathbf{t}$ and 2) the normal of the first intersecting face should be negative. Second, the dot product between 1) a vector from $\mathbf{v}$ to $\mathbf{t}$ and 2) the normal of all $\mathbf{v}$'s neighbor faces should be positive. The second condition is to remove invalid shallow collision to $\mathbf{v}$'s neighbor faces. If any of the two conditions is not satisfied, no collision is detected; hence, return zero loss and terminate the loss computation.

6. Return $\text{loss}(\mathbf{v}) = ||(\mathbf{v} - \mathbf{t})||_1$.

## E    BASELINES

### E.1    VAE

The VAE baseline consists of an encoder and decoder. The encoder of the VAE baseline is almost the same as our denoising network $f$. First, a concatenation of each joint's 3D joint coordinates and 6D rotations of the same handedness is passed to a shared linear layer to change the dimension from 9 to 256. Then, 8 Transformer encoders take $7 + 2J$ tokens, which consist of the output of the linear layer and 7 learnable tokens. Outputs from the learnable tokens are passed to a linear layer to obtain the mean and standard deviation of the data distribution. Using reparameterization trick Kingma & Welling (2014), we randomly sample a point from the estimated mean and standard deviation.

The decoder of the VAE baseline consists of Transformer decoders and linear layers. The sampled point from the reparametrization trick is passed to 8 Transformer decoders. Then, outputs from the same handedness are passed to a shared linear layer, which predicts the 3D coordinate and 6D rotation of each joint.

The VAE baseline is trained with a single A100 GPU, which takes 1 day for the training. We initially set the learning rate to $10^{-4}$, and it decays by 0.1 at $85^{\text{th}}$ and $95^{\text{th}}$ epochs. The training is finished after 100 epochs. During the training, we set the size of the mini-batch to 32. We use the same loss function of the denoising network $f$ to train the VAE baseline. We set the weight of the collision avoidance loss $L_{\text{col}}$ to 0.1. Additionally, we minimize KL divergence between the estimated distribution and a normal Gaussian distribution with weight $10^{-2}$.

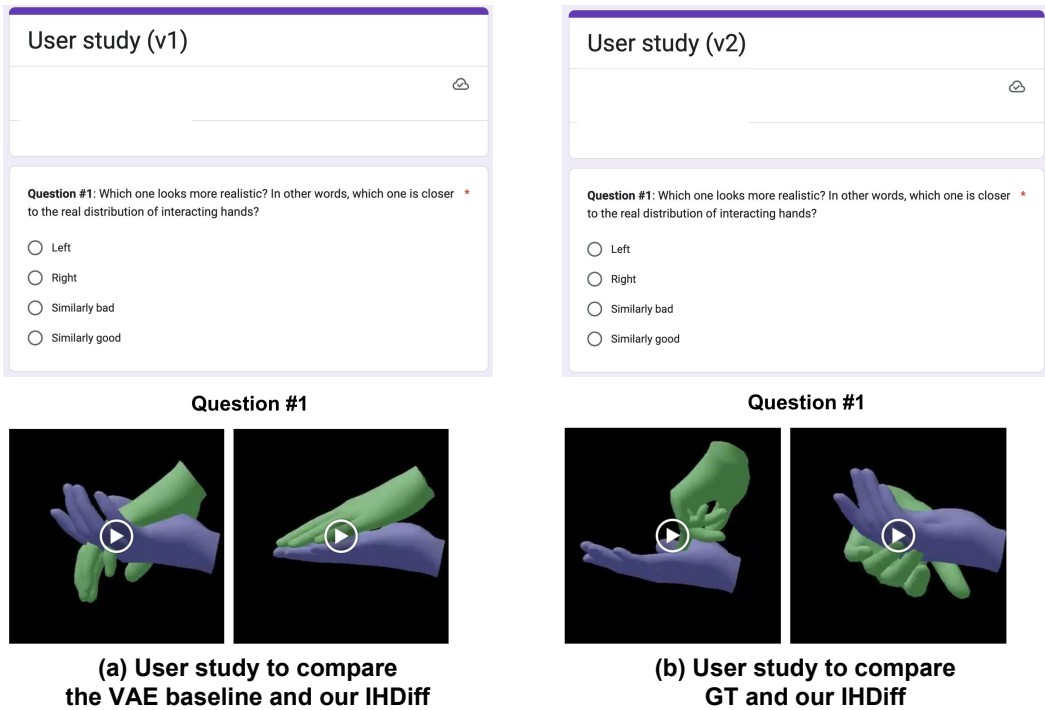

Figure 13: Screenshots of our user studies. For each user study, only one question among 16 questions is shown.

## E.2 OTHER BASELINES

We tried to implement a neural distance fields (Chibane et al., 2020)-based baseline, motivated by Pose-NDF (Tiwari et al., 2022). However, we found that it is not trivial to apply their approach to ours. In their distance fields, a single point is defined as a set of 3D joint angles, and the point is manipulated with a scalar distance and a unit gradient. On the other hand, in our case, a single point should include both 3D angles and 3D translation, where the 3D angles represent the 3D pose of each hand and the 3D translation represents the relative position between two hands. As units of 3D angles (radian) and translation (meter) are not the same, it is not straightforward to define a scalar distance that manipulates both physical values.

## F USER STUDIES

In Sec. 5.2, we conducted two user studies. The first one is to compare 1) randomly generated samples from the VAE baseline and 2) randomly generated samples from our IHDiff. The second one is to compare 1) randomly generated samples from our IHDiff and 2) random GT. Each user study is conducted without noticing comparing methods to participants. Fig. 13 shows screenshots of our user study. We conducted the user studies with a combination of Google Forms and Google Slides. The videos in the Google Slide include rendered interacting hands from rotating viewpoints. Please note that we randomly select the left and right positions of all videos. We attach all videos used for our user studies in 'User study' folder of the supplementary material.

