# OpenReview forum: "3D Interacting Hands Diffusion Model"
_ICLR.cc/2024/Conference — Submitted to ICLR 2024_

### Official Review · Reviewer_MWq4 · 2023-10-28

**Soundness:** 2 fair
**Presentation:** 3 good
**Contribution:** 2 fair
**Rating:** 5
**Confidence:** 5

**Summary:**

This work proposes a denoising diffusion model for generating interacting hand meshes. The method is used for randomly sampling pairs of interacting hands,  conditional sampling by replacing parts of the input with the corresponding hand conditions. It can also act as an post-processing step for 3D hand meshes by fitting to observations.

**Strengths:**

* The proposed method that leverages diffusion models and collision avoidance loss to learn a generative model to capture interacting hand priors is reasonable.
* The proposed method successfully produces reasonable and physically plausible results in three applications, achieving better results than baselines.

**Weaknesses:**

* The DDIM sampling steps are set to 1000, which makes the fitting extremely slow and brings marginal improvement to vertex error.
* The novelty is quite limited. The architecture is not different from the current 3D hand mesh reconstruction methods where cross-attention from different hands is widely used. What makes these three applications useful?
* Predicting the original clean sample instead of predicting added noise for denoising is a common practice in various diffusion model applications.

**Questions:**

* Any network can model the prior distributions of hand meshes. What benefits does the method offer, aside from modeling through a diffusion model?
* Why is generating interacting hand meshes considered an important task?
* Only the VAE part of MLD is used for comparison. More diffusion model-based baselines are needed for comparison.

---

> ### Author Response · Authors · 2023-11-10
> **Rebuttal**
>
> Dear Reviewer MWq4,
>
> Thanks for giving valuable comments.
> Let us address the concerns.
>
> **1. Sampling speed**
>
> Like many diffusion-based methods, we agree that sampling speed could be one of our weaknesses.
> We will clarify this in the final version.
> We observed that even with such a high number of iterations, thanks to our low-dimensional representation of two-hand (201 dimension), our sampling speed is affordable.
> We observed that it takes about 25 seconds for 1000 iterations, shorter than 1 minute of Tevet et. al. (MDM).
>
> **2. Novelty**
>
> We think our biggest novelty is using image-independent generative model to **address the domain gap between in-the-lab datasets (e.g., InterHand2.6M) and in-the-wild images.** As discussed in the Introduction section, 3D annotations are available only for images captured from a lab environment, such as InterHand2.6M dataset, where those images have very different appearances (e.g., background, color space, and illuminations) from those of in-the-wild images. **Hence, simply training networks, including Zuo et al. ICCV 2023 and IntagHand, on such in-the-lab datasets fail to produce good results on in-the-wild images.** We could not experiment with Zuo et al.'s method as their code and pre-trained checkpoint are not available. **IntagHand gives 55.62 mm vertex error, 1.23 collision, and 3.57 contact, which is worse than the InterWild baseline and our IHDiff.** We used their official code and checkpoint. Please note that our numbers of IntagHand could be different from those from InterWild paper as we calculated the vertex error considering 3D relative translation between two hands, while the numbers in InterWild ignored it. Such worse results than InterWild are already verified in InterWild's paper. We think Zou et al. would suffer from the same problem as it does nothing to address the domain gap. On the other hand, our prior distribution is not conditioned on any data including images; hence, **even if it is trained only on in-the-lab datasets, it can be used to improve 3D interacting hands quality from in-the-wild images.** Table 3 and Figure 7 show that our IHDiff improves InterWild in physics-based metrics while preserving the vertex error.
>
> **3. Predicting the original clean sample**
>
> We agree and we do not argue this is our main contribution.
> Please check out the above rebuttal (2. Novelty) for our main contribution.
>
> **4. What benefits does the method offer?**
>
> Please refer the the above rebuttal (2. Novelty).
>
> **5. Why is generating interacting hands meshes considered an important task?**
>
> There could be two benefits.
> First, being able to generate means we successfully modeled the prior distribution of 3D interacting hands. By successfully modeling the prior distribution, we could bridge the domain gap between in-the-lab and in-the-wild environment, as described in (2. Novelty).
> Second, it can be used for many action decision problems, including human-like robots. Let's say human-like robots are doing some jobs or having social interaction with humans. For every timestep, robots should decide what to do next based on current/past observations and status. Robots should not do any unexpected or unnatural behaviors. In this case, we can regularize the next 3D hand pose of robots be included in our prior distribution so that robots do some plausible and meaningful interaction between two hands. Without such a regularizer, robots can do very unnatural hands, for example, collisions between two hands, which can destroy the robots.
>
> **6. More comparison to diffusion-based methods**
>
> Unfortunately, there are no diffusion models for 3D interacting hands.
> Instead, in Table 2, we compared ours with IHDiff (SA only).
> **In Section 3.2 and page 8, we described that the IHDiff (SA only) has almost the same network architecture as Tevet et. al. (MDM).**
> Table 2 shows that our IHDiff produces much better results than IHDiff (SA only).
> We will clarify more about this in the final version.
>
> Please let us know if the reviewer has more concerns or more things to discuss. We are happy to have any discussion.
>
> Best, Authors

---

> > ### Comment · Reviewer_MWq4 · 2023-11-19
> >
> > Dear authors,
> >
> > Thanks for clarification.
> > * Regarding your response to Comment #2: The manuscript seems to identify the primary contribution as a postprocessing step in 3D hand mesh reconstruction. While this is a noteworthy endeavor, the current implementation appears to suffer from limitations in terms of processing speed and offers only marginal enhancements in output quality. This raises concerns about the method's effectiveness and efficiency as a postprocessing tool.
> >
> > * For comment #5, it is commonly acknowledged that the available datasets are inherently limited and do not encapsulate the full distribution of hand interactions. This limitation is also evident in your method. Furthermore, there are concerns regarding the efficacy of sampling poses from a prior distribution, particularly in comparison to the application of more stringent constraints. These constraints could be crucial in regularizing the prediction of subsequent 3D hand poses, especially in robotic applications.

---

> ### Author Response · Authors · 2023-11-19
> **Rebuttal**
>
> Hi Reviewer MWq4,
>
> Thanks for raising valuable concerns.
> Let us address your concerns.
>
> 1. Following Reviewer MWq4's suggestion, we decreased the number of iteration to half, which reduces the fitting time to half. It actually improves the vertex error of Table 3 to 19.63, while achieving almost the same collision (0.88) and contact metrics (12.91). We tried to reduce the iteration even more, but the overall numbers become worse. We agree that the fitting time is one of our limitations and will clarify this in the manuscript. Despite the limitation on the fitting time, it would be greatly appreciated if the reviewer recognizes that our IHDiff is the first unconditional generative model for the 3D interacting hands, and achieves clearly better results than existing regressors and other generative baseline models, such as VAE and diffusion-based MDM-style one (IHDIff (SA only)).
>
>
> 2. The reviewer's concern on the dataset is actually applicable to all data-driven learning-based systems, not only for ours. We are not clear why the reviewer thinks such a limitation is evident in our pipeline, but all experimental results, done on **unseen data**, show how our IHDiff generalizes well to unseen data. For example, Table 2 is performed on InterHand2.6M test set (30 fps), unseen during the training of IHDiff, and Fig. 7 is from unseen in-the-wild video, captured by a mobile phone. Last, Table 3 is from HIC dataset, not used for the training of IHDiff. In particular, InterHand2.6M test set (30 fps) consists of highly diverse 3D interacting hands, while different from those of the training set. We think this makes our evaluation on the generalizability reliable.

---

> > ### Author Response · Authors · 2023-11-20
> > **Rebuttal**
> >
> > Regrading the second concern (generalizability) of the Reviewer MWq4, to further evaluate the generalizability of our IHDiff, we evaluated IHDiff on synthesized errors on top of the test set of InterHand2.6M in Table 3. Our IHDiff can recover the original 3D interacting hands (before adding the synthesized errors) much better than other variants. The experimental results show that our IHDiff is highly robust to unseen errors and can reliably recover errors from 3D interacting hands.

---

> > > ### Comment · Reviewer_MWq4 · 2023-11-22
> > >
> > > Dear authors,
> > >
> > > Thanks for your response. I appreciate the clarifications and efforts made to address some of the issues raised. While I acknowledge the pioneering nature of your work in developing the first unconditional generative model for 3D interacting hands, which is indeed an achievement, I have decided to maintain my original evaluation score. This decision is primarily due to the straightforward methodology employed in the study, which, while effective, leaves some aspects of innovation and depth to be explored further.

---

> ### Author Response · Authors · 2023-11-22
> **Rebuttal**
>
> Thanks for recognizing the significance of our work.
>
>
> Although there is a little technical innovation in the network architecture (cross attention-based Transformer (Ours) vs. self attention-based Transformer (MDM)), **we think our approach of unconditional generative model has strong novelty.** Existing generative models for 3D interacting hands recovery mainly belong to **conditional generative model like Zuo et al. ICCV 2023.** Although their conditional generative model can perform well on studio dataset, such as InterHand2.6M, by being fully supervised by 3D GT, their system fail to produce robust results from in-the-wild images. **This is because in-the-wild images do not provide 3D GT, while their system requires paired (image, 3D GT) for the training; hence, their system suffers from the domain gap between studio and in-the-wild datasets.** Such domain gap is necessary to be addressed, but greatly challenging due to the 1) big image appearance domain gap between studio and in-the-wild images (studio images mainly have black backgrounds with artificial illuminations. Please see images of InterHand2.6M) and 2) difficulty of capturing 3D GT from in-the-wild environment as setting the multi-camera setup in in-the-wild environment requires huge amount of manual efforts and expert-level techniques of camera calibration and synchronization.
>
> On the other hand, **we smartly addressed the domain gap by choosing the unconditional generative model.** Although our unconditional generative model is trained only on studio datasets with 3D GT, it can generalize well to in-the-wild images as **no image appearances are conditioned.**
>
> Hence, **designing unconditional generative model itself is a research problem** and we firstly tried it to address the domain gap using it. It would be great if the reviewer see **the motivation of our approach for the real-world applications instead of focusing on technical details, such as network architectures.**
>
> Thanks,
> Authors

---

### Official Review · Reviewer_uZ7W · 2023-10-30

**Soundness:** 3 good
**Presentation:** 3 good
**Contribution:** 3 good
**Rating:** 6
**Confidence:** 3

**Summary:**

The authors introduce a pioneering generative model tailored for 3D interacting hands. This innovative approach harnesses the power of a diffusion model integrated with a transformer architecture, incorporating both Self-Attention (SA) and Cross-Attention (CA) mechanisms. Notably, the algorithm is versatile, allowing for random sampling, conditional random sampling, and adeptly fitting to observational data.

**Strengths:**

1. Innovative and Practical Algorithm: This study presents the pioneering generative model tailored for 3D interacting hands, characterized by its unique network design. The significance of this prior distribution modeling is underscored by its fitting to observation capabilities, which outperforms current regressor methods both quantitatively and qualitatively.
2. Clarity and Accessibility: The manuscript is exceptionally clear and reader-friendly. Even those unfamiliar with the domain can grasp the content effectively.
3. Thorough Experiments and Analysis: The authors have conducted comprehensive experiments across three distinct settings, showcasing the model's efficacy on both simulated and real datasets. The research is bolstered by user studies, qualitative analyses, and quantitative results, making a compelling case for the model's strengths. The appendix further enriches the paper by detailing the problem settings, network configurations, dataset specifics, and baseline comparisons.

**Weaknesses:**

1. The methodology for integrating the generative prior with the regressor in an end-to-end manner remains ambiguous. It would be beneficial if the authors could delve deeper into this aspect, either in the appendix or the section dedicated to future work.
2. The model's capacity for enhanced generalization, especially concerning varying hand sizes and shapes, is not explicitly addressed. A clearer exposition on this topic would provide valuable insights into the model's adaptability and robustness.

**Questions:**

Please see the weakness above.

---

> ### Author Response · Authors · 2023-11-10
> **Rebuttal**
>
> Dear Reviewer uZ7W,
>
> Thanks for giving valuable comments. Let us address the concerns.
>
> **1. More concrete description of future work**
>
> Once our IHDiff is trained, we can design a regressor that takes a single image and estimates the latent code.
> Then, the latent code can be passed to our pre-trained IHDiff to decode it to 3D interacting hands.
> By estimating additional 3D global orientation and translation of the right hand, we can put the decoded 3D interacting hands in the camera-centered coordinate system.
> Then, we can supervise the 3D interacting hands with various targets, such as 2D joint coordinates.
> While training the regressor, we think there could be two options: fixing the pre-trained IHDiff or fine-tuning it.
> We think this is a research problem.
> If we fix IHDiff, it can preserve useful prior knowledge, learned in the studio data; however, there could be a small domain gap to the in-the-wild datasets.
> On the other hand, if we fine-tune IHDiff, we can further reduce the domain gap; however useful prior knowledge could be diminished.
> We think both directions make sense, and different research contributions should be made for those two directions.
>
> **2. Generalization on the varying hand sizes and shapes**
>
> Thanks for pointing out such a valuable comment.
> We actually agree with the reviewer.
> In the current manuscript, what we did for the hand shape is regularizing it with our collision avoidance loss.
> This should encourage our IHDiff to give hand shape parameters that do not introduce collisions given 3D poses; however, it is not fully investigated in terms of *diverse* hand shapes.
> For example, the extremely thick hand could still introduce collision between two hands.
> For this research direction, we think we first need to collect a 3D hand dataset with diverse hand shapes as currently available ones do not have enough diversity.
> We'd like to push that research direction in the future.
>
> We will include and clarify all valuable comments and suggestions from the reviewer.
> If the reviewer has more things to discuss, please let us know.
> We are happy to have more discussion.
>
> Best,
> Authors

---

### Official Review · Reviewer_Z2YY · 2023-10-31

**Soundness:** 3 good
**Presentation:** 3 good
**Contribution:** 1 poor
**Rating:** 5
**Confidence:** 5

**Summary:**

Authors proposed to apply the diffusion model for generating 3D interacting hands. This is the first attempt to learn prior distributions of interacting two hands in 3D space. They leverage transformer-based architecture as the denoising network to better capture inter-hand interactions. Experiments demonstrates the SOTA results on interhands2.6M dataset in terms of vertex error and contact accuracy.

**Strengths:**

Diffusion model was first used for interacting hand pose estimation task; while some were proposed for human body domains.
Achieves SOTA results on handling contacts and collisions
The work claims the proposed collision loss can prevent both self-collisions and inter-collisions while the widely used (SDF)-based collision avoidance loss function can only handle the inter-collisions.
The english is well written.
Throughout diverse experiments, authors proves the effectiveness of the diffusion model in interacting hand domain.

**Weaknesses:**

The work fails to achieves SOTA in the in terms of vertex error.
Technical contribution seems weak. The only contribution seems like that using transformer-based architecture for denoising process.
Simple methods such as nearest neighbor search outperforms their method in the ratio of collision. This makes it unconvincing for their results.

**Questions:**

Authors proposed the right hand-relative space to represent inputs. I’m curious if it is possible to represent inputs based on the left hand relative space.
The work claims to model the priors for interacting hands. However, I am curious about the generalization capability of the model. How effectively the prior can be applied to the unseen data?

---

> ### Author Response · Authors · 2023-11-10
> **Rebuttal**
>
> Dear Reviewer Z2YY,
>
> Thanks for giving valuable comments. Let us address the concerns.
>
> **1. Vertex error**
>
> Please note that we achieved almost the same vertex error (0.7 mm difference) compared to the state-of-the-art method while significantly improving the physics-based metrics, such as contact and collisions. In particular, contact between two hands is critical to determine semantic meaning of interaction between two hands. Ours is the first work that explicitly address the contact metric between two hands.
>
> **2. Contribution**
>
> We think our biggest novelty is using image-independent generative model to **address the domain gap between in-the-lab datasets (e.g., InterHand2.6M) and in-the-wild images.** As discussed in the Introduction section, 3D annotations are available only for images captured from a lab environment, such as InterHand2.6M dataset, where those images have very different appearances (e.g., background, color space, and illuminations) from those of in-the-wild images. **Hence, simply training networks, including Zuo et al. ICCV 2023 and IntagHand, on such in-the-lab datasets fail to produce good results on in-the-wild images.** We could not experiment with Zuo et al.'s method as their code and pre-trained checkpoint are not available. **IntagHand gives 55.62 mm vertex error, 1.23 collision, and 3.57 contact, which is worse than the InterWild baseline and our IHDiff.** We used their official code and checkpoint. Please note that our numbers of IntagHand could be different from those from InterWild paper as we calculated the vertex error considering 3D relative translation between two hands, while the numbers in InterWild ignored it. Such worse results than InterWild are already verified in InterWild's paper. We think Zou et al. would suffer from the same problem as it does nothing to address the domain gap. On the other hand, our prior distribution is not conditioned on any data including images; hence, **even if it is trained only on in-the-lab datasets, it can be used to improve 3D interacting hands quality from in-the-wild images.** Table 3 and Figure 7 show that our IHDiff improves InterWild in physics-based metrics while preserving the vertex error.
>
> **3. Comparison to NN**
>
> Please note that **NN retrieves the nearest GT from the training set for each sample of the test set**. The ratio of collision is computed only using the output of each method (retrieved GT for NN's case), not against GT from the testing set. Hence, by the nature of GT, it gives low ratio of collisions. In other word, the ratio of collision checks the quality of the output (retrieved GT for NN's case) and say nothing about the similarity to the GT of the test set. On the other hand, it gives low vertex error and contact accuracy as these two metrics measure similarity to GT from the testing set.
>
> **4. Left hand-relative space**
>
> We think the model can also learn left hand-relative space. Actually, the left hand-relative space can be easily modeled by appying the inverse transformation of the root joint of the left hand to our representation.
>
> **5. Generalization to unseen data**
>
> All the numbers in Table 2 and Table 3 are from unseen test data.
> In addition, all results of Figure 7, Figure 10, Figure 11, and Figure 12 are from unseen test data.
>
> Please let us know if the reviewer has more concerns or more things to discuss. We are happy to have any discussion.
>
> Best, Authors

---

> > ### Comment · Reviewer_Z2YY · 2023-11-19
> > **Comment on author feedback**
> >
> > I think author's explanation regarding the poor accuracy seems not be the proper answer to my comments, saying the gap between the previous paper's number and their own re-reproduced results. I can only confirm that the performance is not completely the state-of-the-art.
> >
> > I newly realized that the results are all obtained for unseen test data, and this seems to be an interesting results. However, I still think the technical contribution is weak. In my opinion, the contact between two hands is also not that novel contribution in the context of hand pose estimation. There seems less additional component to the diffusion model.

---

> ### Author Response · Authors · 2023-11-19
> **Rebuttal**
>
> Hi Reviewer Z2YY,
>
> We think there is a misunderstanding. The results of IntagHand are not from our own reproduced ones, but as we said, **they are from their official code and pre-trained checkpoint.**
> In other words, we used code and pre-trained checkpoint from here: https://github.com/Dw1010/IntagHand. **The different numbers are only from a different evaluation protocol** because previous MPVPE calculates the vertex error for each right and left hand separately by normalizing the translation of each hand with each root joint. **This is exactly the same way of calculating the error for the single hand case, which cannot consider interactions between two hands.** On the other hand, we calculate the vertex error by normalizing the translation of the two hands with the right hand root joint, which means the relative translation between two hands is not canceled during our evaluation. We think **ours is a better metric because the relative translation between two hands is an important signal to define interactions between two hands, such as contact.**
>
>
> In terms of collision and contact between two hands, our results are clearly state-of-the-art. We improved the contact accuracy **100%** compared to InterWild, and  **400%** compared to IntagHand. Figure 7 shows that collision and contact between two hands are necessary metrics. In particular, compared to InterWild, **the positions of two hands (vertex error) is similar, but ours 'looks' much better in Figure 7 because of the less collision and better contact.**
>
> In addition, we decreased the number of iteration to half to reduce the fitting time to half. It actually improves the vertex error of Table 3 to 19.63, while achieving almost the same collision (0.88) and contact metrics (12.91), **which achieves state-of-the-art results on all metrics.**

---

### Official Review · Reviewer_h3k4 · 2023-10-31

**Soundness:** 3 good
**Presentation:** 3 good
**Contribution:** 3 good
**Rating:** 6
**Confidence:** 4

**Summary:**

This paper provides a diffusion based two-hand interaction prior. This prior enables many applications such as random pose sampling, conditional pose sampling and image pose fitting. The core algorithm is a Transformer based network that encode the left-right-hand interaction through cross-attention.

**Strengths:**

1. This is the first two-hand interaction prior using diffusion model. Several applications have been enabled.

2. The results are impressive, especially the physical plausibility compared with VAE baseline.

**Weaknesses:**

1. The method seems to be a trivial modification of diffusion process with limited novelty.

2. The evaluation is not satisfactory, see questions below. A prior model is definitely useful for hand pose sampling or hand pose estimation. However, more thorough comparisons are necessary to convince me with the effectiveness of this prior on single-image two-hand recovery.

**Questions:**

My concerns mainly relate to the comparison for single image hand reconstruction. The paper mainly compared with InterWild (Moon et.al. CVPR 2023) in Table.3 on HIC dataset. (1) Why not use interhand2.6M dataset for comparison? (2) I noticed that the authors cited "Reconstructing Interacting Hands with Interaction Prior from Monocular Images" (Zuo et.al. ICCV 2023), which is the new sota. It would be better to compare with it as it also claims an interaction prior. Although Zuo et.al. handled a more narrow task (image based hand reconstruction), it is valuable to discuss the difference of the two priors for image-based hand reconstruction.  (3) I would like to see a direct comparison with IntagHand (Li et.al. CVPR 2022) for single-image two-hand recovery. In my opinion, IntagHand focused on capturing the interaction part of hands using attention, which might be more suitable than InterWild as a baseline.

---

> ### Author Response · Authors · 2023-11-10
> **Rebuttal**
>
> Dear Reviewer h3k4,
>
> Thanks for giving valuable comments.
> Let us address the concerns.
>
> **1. Novelty**
>
> We think our biggest novelty is using image-independent generative model to **address the domain gap between in-the-lab datasets (e.g., InterHand2.6M) and in-the-wild images.** As discussed in the Introduction section, 3D annotations are available only for images captured from a lab environment, such as InterHand2.6M dataset, where those images have very different appearances (e.g., background, color space, and illuminations) from those of in-the-wild images. **Hence, simply training networks, including Zuo et al. ICCV 2023 and IntagHand, on such in-the-lab datasets fail to produce good results on in-the-wild images.** We could not experiment with Zuo et al.'s method as their code and pre-trained checkpoint are not available. **IntagHand gives 55.62 mm vertex error, 1.23 collision, and 3.57 contact, which is worse than the InterWild baseline and our IHDiff.** We used their official code and checkpoint. Please note that our numbers of IntagHand could be different from those from InterWild paper as we calculated the vertex error considering 3D relative translation between two hands, while the numbers in InterWild ignored it. Such worse results than InterWild are already verified in InterWild's paper. We think Zou et al. would suffer from the same problem as it does nothing to address the domain gap. On the other hand, our prior distribution is not conditioned on any data including images; hence, **even if it is trained only on in-the-lab datasets, it can be used to improve 3D interacting hands quality from in-the-wild images.** Table 3 and Figure 7 show that our IHDiff improves InterWild in physics-based metrics while preserving the vertex error.
>
> **2. Why not use InterHand2.6M for Table 3?**
>
> As described in the paragraph right above Section 6, networks trained on InterHand2.6M already perform well on test images of InterHand2.6M as training and testing sets of InterHand2.6M look almost the same. As we described in the above rebuttal (1. Novelty), we are focusing on more real-world applications by **addressing the domain gap between in-the-lab datasets and in-the-wild images**. Hence, we used the HIC dataset (Table 3) and Figure 7, which consists of images captured in a daily environment, not from a studio.
>
> Please let us know if the reviewer has more concerns or more things to discuss.
> We are happy to have any discussion.
>
> Best,
> Authors

---

> > ### Comment · Reviewer_h3k4 · 2023-11-19
> > **Comment to authors feedback**
> >
> > I would like to maintain my original rating. I think the domain generalization ability is vital for real-world applications. The paper proposes a direct yet effective (seems to be) prior of two-hand interacting, which may help improve real-world hand mocap quality. I do not lift my rating because the idea is truly straightforward.

---

> > > ### Author Response · Authors · 2023-11-19
> > >
> > > Hi Reviewer h3k4,
> > >
> > > Thanks.
> > > Please let us know if you have more things to discuss.

---

### Meta-Review · Area_Chair_qau7 · 2023-12-07

**Metareview:**

This work proposes a technique for generating 3D interacting hands. Reviewers appreciated good results. Multiple reviewers raised concerns regarding insufficient experiments. Area chair carefully read all the reviewer and author messages and agree with the reviewer assessments. Authors partially addressed some of the concerns in their rebuttal. But, it is felt that the paper needs to be improved to fix several issues. Area chair discussed the strenghts and weaknesses of the paper with the reviewers and all the reviewers felt that the paper is not yet ready for the publication at ICLR. The authors are encouraged to consider the reviewers' comments when revising the paper for submission elsewhere.

**Justification For Why Not Higher Score:**

Reviews are leaning towards rejection in both the reviews and discussion. Several issues remain in the paper such as some missing experiments.

**Justification For Why Not Lower Score:**

Reject is lowest score.

---

### Decision · Program_Chairs · 2024-01-16

Reject